# Microbiome Research: Open Communication Today, Microbiome Applications in the Future

**DOI:** 10.3390/microorganisms8121960

**Published:** 2020-12-10

**Authors:** Bettina Schelkle, Quentin Galland

**Affiliations:** 1European Food Information Council, Rue des Deux Eglises 14, 1000 Brussels, Belgium; 2Hague Corporate Affairs, Rue Belliard 40, 1040 Brussels, Belgium; galland@hague.company

**Keywords:** microbiome, communication, trust, innovation

## Abstract

Microbiome research has recently gained centre-stage in both basic science and translational applications, yet researchers often feel that public communication about its potential overpromises. This manuscript aims to share a perspective on how scientists can engage in more open, ethical and transparent communication using an ongoing research project on food systems microbiomes as a case study. Concrete examples of strategically planned communication efforts are outlined, which aim to inspire and empower other researchers. Finally, we conclude with a discussion on the benefits of open and transparent communication from early-on in innovation pathways, mainly increasing trust in scientific processes and thus paving the way to achieving societal milestones such as the UN Sustainable Development Goals and the EU Green Deal.

## 1. Background: Why Do We Need to Communicate Microbiome Research Better?

Microbial communities exist in many different environments across the whole food ecosystem: near food, fodder and forage crop roots in soil, on plants and algae, on fruits and vegetables, in some foods that have been fermented, baked or brewed, on and in animals (including fish and molluscs) and humans, in aqueous environments where food is grown or harvested [1,2]. Understanding the microbiome within and across these different environments is considered the next big breakthrough in science. It will potentially have ripple effects in many other scientific disciplines beyond microbiology and will impact society as a whole, down to the individual level [3,4]. Not surprisingly its vast potential has caught the interest of the general public, food chain actors and policy makers, creating renewed hope for solutions that could tackle global societal challenges such as obesity and climate change [3,5]. However, there is some concern that the microbiome is oversold on its current premise, which may lead to a loss of trust in and a fear of microbial applications by the general public [6,7]. For scientific breakthroughs to occur, it requires the acceptance of new technologies and applications resulting from increased knowledge in any given field by consumers and citizens (including those who are involved in relevant decision making on the microbiome, e.g., for new policies or regulations that ensure food safety) [8]. Some innovations or new scientific knowledge may even require citizens to change behaviours in the interest of individual and public health [9,10]. However, new knowledge can feel threatening to some communities. Their engagement and understanding of the microbiome and its potential application is, thus, crucial in improving public and environmental health. In an ideal world, citizens are empowered with relevant knowledge through successful science communication for informed professional and personal decision making [3,11,12].

Yet the large amount of public “education” on the microbiome currently happens in the press and on social media, with the main focus being the human gut microbiome [6]. Further, it is frequently biased and incomplete, often unknowingly by the authors due to their lack of knowledge of the broader picture. The huge complexity of the topic even within one research sector (e.g., [13,14]) further increases the risk for miscommunication. Communications overselling the microbiome have the potential to undermine trust in those people on which science is reliant on to accept innovative microbiome applications (e.g., [8]), such as farmers using biobased pesticides or consumers buying produce grown whilst biobased pesticides were applied. When a large acceptance of technological applications and refined procedures is not achieved within broader society, breakthroughs in how systems (such as food systems) are managed cannot occur [15].

Effective communication on the microbiome can: 1. increase and maintain confidence and trust of world citizens in scientists; 2. contribute towards social equity and public health; 3. increase transparency in scientific and policy decision-making processes and representativeness; as well as 4. motivate and inspire people “to do their bit” because they clearly see how it links to their personal values (e.g., in support of the urgently needed food system transformation) [16,17,18,19,20,21]. It can prevent misinterpretation of findings, broaden the microbiome narrative from gut health to “One Health” approaches that span across silos, influence public perception and practices, accurately portray the limitations of research findings and how this affects how knowledge is applied in real-world contexts [6,18,22]. When science communication accurately portrays the benefits—and the limitations—of a research breakthrough, it can also improve the quality of policy making and regulation thereby increasing the overall legitimacy of the decision-making process [9,10,11,23] due to a better understanding of the risks and hazards associated with new microbiome applications. The latter will help support citizens and decision makers judge how warranted societal anxieties and fears are, such as those that have been raised by the communications on GMOs and CRISPR–CAS technologies [24,25,26,27]. Finally, it can contribute towards goals outlined in international strategies such as those within the European Union’s Farm-to-Fork strategy and the UN Sustainable Development Goals [28,29].

Calls for proactive expectation management outlining current limitations for potential microbiome application users are becoming urgent to prevent a later backlash from society [3,6,7]. Proactive communication management requires early, clear and open communication on not just potential benefits, but also limitations in ongoing research studies that may affect how microbiome applications can be used in practice [17,19]. Within the research fields of microbiomes such communication proves to be challenging for the following reasons:

The majority of public knowledge on microbiomes is limited to clinical benefits for human health through dietary changes [30,31].

Microbiomes in nonhuman environments have been significantly less exposed to the public eye (such as soils, plants, animals, food waste degradation etc. [3,4,5]) and as such their importance in mitigating climate change [1], improving agriculture productivity [14] and enhancing biodegradation processes [32] is often overlooked.

Microbiomes affect their host environment directly and indirectly, meaning clear causalities between certain external factors and microbiome functions (including their impact on host environment) are difficult to discern [3,4].

There is often a high degree of uncertainty associated with research outcomes as microbiomes are subject to multiple drivers of their functions (e.g., [13]).

Research limitations and their impact on the practical applications of microbiome knowledge are not actively communicated.

A further consideration for the current status quo on communication around microbiomes includes the recognition that societal acceptance of innovative microbial application is not just dependent on the technical knowledge that scientists have on a given subject, but that it is also aligned with personal values and politics of citizens (i.e., recipients of scientist’s communications’ efforts) [11,33]. As such, interdisciplinary approaches between microbiologists, behavioural scientists and sociologists may be needed to clearly understand the intersection between what people need to know, rather than what may appeal to the public or what scientists want to disclose in public engagement efforts, to achieve public support for microbiome applications [4,12,23,34]. Most importantly, communication with the lay public will need to centre on an approach that allows for a dialogue and engagement to gain the necessary support for microbiome applications rather than the classic, one-way transmission of knowledge, which is slowly becoming an outdated and ineffective approach [35,36,37].

Whilst Timmis et al.’s [3] proposal of building microbiome literacy through school education is the ideal for tackling the situation whereby the microbiome may be oversold in its premise, scientists cannot await a new generation to come into decision-making powers. Instead, there is an acknowledgement that we have to work with the current population at large to bring them onboard and provide them with the opportunity to deliberate and decide with us scientists and science communicators on microbiome applications, rather than being bystanders apart from the decision-making process [19]. Such approaches are being implemented in citizen science projects where citizens support the collection of data under the guidance of established scientists to ensure scientific principles and methods are robust [38]. Citizen science projects within the microbiome field include an initiative by the French Institute for Agriculture and Environment to build a publicly available gut microbiome database (e.g., [39]).

The aim of the current paper is to outline a perspective on how a proactive communication approach can appear within the limitations of a standard research project, using the example of the H2020-funded Innovation Action project CIRCLES (Controlling mIcRobiome CircuLations for bEtter food Systems, www.circles-project.eu), to enhance trust and acceptance of future microbiome applications. For the purpose of this manuscript, we consider the microbiome within the food system rather than human gut health context, although the applicability of some of our messages and discussions are inclusive of both reference frames.

## 2. How Can We Effectively Communicate About the Microbiome?

Effective communication on any topic requires a tailor-made strategy, which clearly addresses and defines the objectives of the communication, the target audience, the messages, as well as the channels one intends to use to convey their messages. While communication processes have previously been one-way oriented [40], the rise of social behavioural sciences [41,42] and the increasing use of social media [43] have led to increased interaction with the target audience. The involvement of the audience has encouraged communicators to rethink their strategies, paving the way for sustained interactions with their target audience.

In this manuscript, we use the example of CIRCLES to outline a clear and dynamic communication strategy evolving alongside project needs and outcomes. The project aims at understanding how members of the microbiome interact with one another across environments in the food system and, in turn, how this influences the production of vegetables and fruits, or the production and wellbeing of livestock and wild fishes. Such an understanding will identify weak spots of production systems, where microbiome applications (e.g., probiotics for soils) could be developed that result in win–win solutions for the current pressures the food system faces (e.g., biodiversity decline, antimicrobial resistance).

As part of an effective communication strategy, the project partners early-on defined some general principles, including the purpose of communication, key messages (see Box 1), the target audience, methods of communication and timing (for practical examples see Table 1).

Box 1Target audiences and key messages of the CIRCLES project.**Consumers**: Microorganisms are an inseparable part of our environment, even though they are largely invisible. These organisms forming the microbiome live in and around us. Scientists study their impact on our health and lives and, therefore, assess how to use them to improve our food production and our health.**Policymakers**: Growing world population, risks of food scarcity and the impact of climate change drive the impetus for sustainable food production. New applications based on microbiome research can support the deployment of sustainable food systems. Policymakers can contribute to such a deployment by staying abreast with research and enabling evidence-based regulation.**Farmers and food growers**: Microbiome research offers the chance to support farmers with farming applications that could increase yields while reducing the use of pesticides and chemical fertilisers and maintaining or improving animal and plant health. In the project CIRCLES (Controlling mIcRobiome CircuLations for bEtter food Systems), we are assessing how microbiomes impact food production and whether these could lead to a new generation of safe, high-quality and affordable feed, food additives or new biopesticides.**Food and feed manufacturers**: Research on microbiomes can lead to a new generation of food and feed products that would maintain the competitive edge of companies whilst ensuring the sustainability and safety of their products.**Journalists**: CIRCLES performs microbiome research and explores microbiome dynamics and circulation through laboratories in the fields. The project’s outcomes may lead to innovative farming methods relying on a holistic view of the food system where microbes play a central role and that could further pave the way towards healthier food products and a sustainable agricultural sector.

### 2.1. Purpose of Communication

The purpose of each project within the European Commission’s Horizon 2020 framework (https://ec.europa.eu/programmes/horizon2020/en) is to deliver impact. CIRCLES specifically aims to support the urgently needed food system transformation by exploring the potential of microbiomes across the food chain to deliver highly nutritious foods in a sustainable way using the One Health approach. Hence, the communication aims are to:

Raise awareness of what microbiomes can and cannot do for a more sustainable food system and human health;

Highlight regulatory hurdles that may hold back microbiome innovation from reaching markets;

Share knowledge with food chain actors that can utilise microbiome innovations to make production processes more sustainable and efficient.

### 2.2. Target Audience

To deliver impact, project participants clearly need to understand who will use their research outputs and who may be indirectly affected by these outcomes. Establishing and assessing the interest of different audience segments is crucial. For example, a journalist would most likely be interested in learning about the outcomes of the project work, the challenges encountered and the significance of the research for society; a farmer, on the other hand, would be interested in knowing how microbiome applications can be used safely and cost-effectively for growing foods in alignment with their ongoing farming practices [17,44]. Citizens are interested in the microbiome for personal health applications. Further breakdowns within a specific target group may also be required. 

### 2.3. Target Messages

As a key principle, keep it (your messages) short and simple (KISS) and relevant to the specific target audience (see Box 1). Focus on what the target audiences are interested in and clearly communicate the potential limitations of the research and its applications. Technical details and approaches usually discussed in depth with scientific colleagues or for evidence-based policy making are, for instance, not necessary in communicating with the general public. Instead, appeal to the target audiences’ interests by using story-telling approaches, analogies and terminology they are familiar with. Story telling approaches have a twofold benefit. First, stories aim to provide tangible examples for demystifying the world of microbiomes. Second, such stories aim to bring science closer to people by facilitating access to behind-the-scenes activities. Complete the messages you want to convey to the lay audience by utilising channel-specific tools, such as graphics within presentations, and hashtags and emojis for social media platforms, for more effective messaging. 

### 2.4. Communication Channels

To make sure the messages reach their target audience, they need to be disseminated through the appropriate communication channels. Whilst the scientific literature and exchange at conferences is important for science development, engagement with the lay audience usually takes place outside of traditional science channels and includes the press and social media. For instance, farmers may be best reached through national farming magazines published in relevant languages on farming practices that could be improved through microbiome applications, citizens aged 18–34 and concerned about their food produce are likely on social media (specifically Instagram and Facebook) and fellow scientists can be reached at relevant conferences through oral and poster presentations. 

Besides the conventional face-to-face exchange of information, communication channels have progressively diversified with the advent of the Internet, which has the advantage of having unparalleled reach. Indeed, younger Europeans prefer to get their news online [45]; hence, the prerequisite for microbiome communication to capitalise on online channels (e.g., social media, website) as well as engaging in more traditional communication methods. 

### 2.5. Timing

The increasing curiosity of the general public towards the microbiome presents a window of opportunity for researchers to further spark the interest of the audience. The impact of messages can be enhanced by communicating them through other significant communication channels. For instance, a press release on a project final conference could coincide with the publication of the major result for a project combined with a paid social media campaign. In addition, preparing and publishing communication activities with already existing microbiome communication campaigns provide an additional stimulus to the communication outreach. The main advantage is that a recently launched microbiome research project can benefit from an already established community base, which will be inclined to learn more about its developments and potentially spread its messages via its own network or channels. World Microbiome Day (www.worldmicrobiomeday.com) and International Microorganism Day (www.internationalmicroorganismday.org/) are good examples that support specific online campaigns with messages providing explanatory information on the microbiome alongside insights and interesting facts. Such days, celebrated worldwide, provide an opportunity for different microbiome research projects to align their communication and increase the scope and impact of the outreach. More recently, the ongoing COVID-19 pandemic has provided an opportunity to outline the importance of viruses within microbiomes, thereby capitalising on a current topic.

## 3. The Importance of Communication Support and Alignment

Science communication has been growing in relevance in the last decade with some academics having taken to specific platforms or building a network amongst science journalists. Yet not all academics feel confident enough to engage consistently in communication efforts beyond presentations at scientific conferences, despite tangible benefits (e.g., expanded network, increased citation rates, [46]). From a project communication perspective, messages can gain additional weight and credibility when communicated by a scientific expert. This is why researchers should proactively take part in the communication activities of the project. To support researchers who have little expertise in communication, the communication lead should provide guidance in different ways. This could consist of communication trainings focused on a specific activity (e.g., how to write a blog article; a social media post) where researchers can engage in hands-on practice and receive feedback.

Another approach to support scientists in communications is the development of ready-made social media messages which they could use in a semiautomated way by copy-pasting and posting them in their professional or personal social media account. A good example used in the CIRCLES project is the provision of social media kits by the communication lead (https://circlesproject.eu/social-media-kit/; Appendix A). 

## 4. Challenges on Engaging with the Wider Public

When engaging with the general public, the expectation should be that, at best, only the segment of the lay audience which is interested in science will be reached. If the overall communication and dissemination is effective, a specific niche audience interested in the microbiome will be addressed (or several, if means for the use of different tools and approaches are available [16]).

Ideally, resources should be allocated to explore what type of communication is needed: where is that sweet spot between what the audience is interested in and what needs to be communicated to gain public support for microbiome applications (rather than what researchers want to communicate [4])? An easy method of obtaining more information about what audiences are interested in is to perform a key word search on frequently asked questions on Google (on the microbiome topic the most searched for result in Google is “what is the microbiome?”). The results of such a key word search can guide the creation of content for project websites, the press or social media, whereby one can take into account the types of questions people are asking online. It provides an ideal hook to follow by answering their question with important points that the scientists want to convey about their specific work.

Once the audience is interested in the overall topic, keep the project-related information short and sweet. In today’s world, most citizens have a short attention span and do not necessarily use an information-seeking approach to find information: instead, heuristic decision making on whether to engage with the content is the norm [47]. Further, if the audience engages with the content that is provided, researchers should be beware that citizens will come with some knowledge already; this knowledge is shaped by personal experience, culture or conventional wisdom [12,33,47]. Hence, researchers may receive questions that are related to the big picture of the topic that is being worked on, but in actual fact may not be directly relevant to the specific project topics themselves. 

Further, scientists are encouraged to clearly communicate about uncertainties and limitations within their research, including gaps and divergent views, as these have important implications for decision making and building trust [19,48,49]. As part of communicating uncertainties, microbiologists also need to outline the path of evidence from studies conducted in the laboratory to high-quality microbiome products.

Finally, monitoring communication efforts to assess impact and adjust ongoing efforts in communicating with different stakeholder groups such as the general public is highly recommended through proactive website and social media management and engagement, the use of Google analytics, joining public fora for scientific discourse and requesting feedback from participants in joining public engagement activities [9,35]. 

## 5. Benefits of Science Communication to Researchers

An active engagement in science communication by researchers that goes beyond scientific publications and conference presentations is starting to provide tangible positive outcomes for individuals. For instance, “tweetations” (scientific publications that are being discussed on Twitter) with a positive sentiment can increase downloads and view metrics of scientific publications in the short term and citation rate in the long term [50,51].

”Tweetations” further appear to negate gender inequalities [52]. Indeed, social media influence of researchers is under consideration for use as a criterion to assign funding to research [53], whilst in the meantime, some funding agencies are providing prizes for the most effective and original approaches for science communication and engagement (e.g., the European Research Council Public Engagement and Research Awards) [54]. Finally, anecdotes from researchers engaging in discussion with the wider public also report that it provides them with good ideas for future research.

## 6. Conclusions

Scientists can gain much by engaging early-on in open, ethical and transparent communication with stakeholders. The aim is to keep people engaged and informed from the earliest stages to come to a win-win for all: 

Consumers know and trust their food and the technologies that are applied to them (including food safety aspects);

Citizens know where to go for information and are empowered to make informed choices;

Farmers and industry understand how to use microbiome applications and are willing to use upcoming products responsibly;

Regulatory efforts support a sustainable, safe food supply;

Scientists gain visibility for their work and deliver real-world impact that contributes to tackling global challenges like climate change, malnutrition and unsustainable food chains.

## Figures and Tables

**Table 1 microorganisms-08-01960-t001:** Example of some of the CIRCLES (Controlling mIcRobiome CircuLations for bEtter food Systems) project’s main aims for communication and dissemination: How are key messages shaped? To whom are they targeted? How and when are they communicated?

Purpose (Why?)	What?	To Whom?	How? *	When?
Raise awareness on microbiomes across the food system and promote engagement	Microbial communities exist in many different environments: near roots in soil, on plants and algae, on fruits and vegetables, in some foods that have been fermented, baked or brewed, on and in animals (incl. fish and molluscs) and humans, in aqueous environments where food is grown or harvested	The general public interested in the microbiome (for personal health or sustainable food system reasons)	Contribute to existing public engagement events such as World Microbiome Day Stands at open science events across Europe where people can test their skin microbiome & speak to researchers about microbes (CIRCLES City Tours) Project Instagram channel	Anytime throughout the project in alignment with other public engagement activities by other researchers (such as on World Microbiome Day and International Microorganism Day)
Raise awareness on potential microbiome products (to allow for dissemination and engagement)	Product details (incl. benefits and limitations) Foreseen regulatory hurdles to bring this product to market	Funding bodies Regulatory bodies Industries/Small-and Medium-sized Enterprises with similar products who may be interested in outcomes Farmers	Press releases and articles that target bioeconomy and specialised magazines (e.g., agricultural) Trade fair attendances Small ad hoc meetings in closed circles (e.g., with farming groups) Project Twitter and LinkedIn channels Project newsletter service	As relevant results become available throughout the project (LinkedIn channel to be established from 2022 onwards to allow for a platform that promotes engagement and interaction with professional stakeholders)
Promote the project itself (mainly for dissemination purposes)	Basic project information, regular updates on progress, outputs (publications, events etc.)	Microbiologists and other scientists Industries/Small-and Medium-sized Enterprises	Website Flyers Scientific publications and conferences Project Twitter channel Project newsletter service	Anytime throughout the project

* Some of the channels overlap for different key messages; for the purpose of simplifying the table we have focused on the main channels only.

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
