# Peer review of "Microbiome Research: Open Communication Today, Microbiome Applications in the Future"

_microorganisms, 2020, doi:10.3390/microorganisms8121960_

Round 1

Reviewer 1 Report

The manuscript by Schelke and Galland is a perspective piece on the shortcomings and possibilities of scientific communication in the microbiome field., presented from the view of the CIRCLES research consortium. The authors argue that a more open and transparent communication of research results on microbiomes in food production and potential applications derived from it will resolve unwarranted fears and increase acceptance of such products in different non-scientific actors, such as farmers, the general public and policy makers. Although this argument is not exactly new, the manuscript provides a good resource for other scientist and also potential non-scientific readers (e.g., journalists) on what to consider when communicating science to the general public and offers a guideline on how to engage in this process.  

Line 43 and elsewhere: The phrase “silo” seems odd in this context and is (to the knowledge of this reviewer) not a common scientific phrase. Please rephrase.

Line 60 and following: This sentence should be rephrased. The aim of open and transparent scientific communication should not be to completely prevent the recipients to be anxious or fearful of novel technologies, but rather to allow them to better understand how eligible or unwarranted their causes of fear actually are and to enable them to communicate these reasons accordingly. GMO- and CRISPR-CAS- (and possibly microbiome-) based technologies are by no means all completely save and reasons to be anxious about their wide-spread application are valid.

Line 73: Rephrase “are have”

Line 118: Remove “microbial”

Line 126 (Box1): “Microbes” and “micro-organisms” are the same thing. “Remain” should be replaced with “are largely” since some of them are visible with the naked eye. Please add an “-s” to “asses”

Line 225: The statement “world citizens have a short attention span” is way to general and probably not true for many “world citizens”. Please rephrase

Author Response

Please, see attachement.

Reviewer 2 Report

General comments:

On this perspective manuscript, the authors claim that an improved system of microbiome science divulgation is needed in order to increase the benefits of microbiome research to society, which is valid. The reasons are properly discussed and the proposed methods are interesting. The quality of writing is in general good, despite some minor grammar edits were suggested. The authors should be cautious when stating that this will cause that. These are hypothetical suggestions and need to be stated as so. Moreover, some statements need to be changed to clarify that the proposed methods of science divulgation to reach the general public refer to alternative media, and acknowledge that the current method of scientific literature is important and useful for science development. Philosophical and basic science questions that are present in microbiome research are essential and the main fuel for scientific discovery. However, a more practical focus using palatable language would be useful in the proposed alternative media to increase impact of microbiome applications in the society. Finally, the benefits for different sectors of the society were discussed, but this was not explored for scientists. How the academic and research institutions could stimulate scientists to engage in scientific divulgation in alternative media should be more discussed, perhaps in an entire topic. Scientists need practical benefits for dedicating time on this instead of generating more results for conventional scientific publications.

Specific comments:

Line 21: The use of “food chain” may be confusing here. In the ecological context, microbial communities are part of the food chain (decomposers). Please clarify the meaning.

Line 22: fodder are also crops, please correct. The best way would be “near food, fodder and forage crop roots in soil (rhizosphere), on and in plants…”. In addition, free-living microbial communities inhabiting soil, water, and other non-host environments were neglected. These microbiomes are present in agroecosystems and eventually contribute to the food systems.

Line 23: Microbiomes are not only observed in human/animals skin and digestive systems, but in virtually all organs – despite the abundance and maybe importance is higher in the digestive systems.

Lines 21-23: References should be used for these sentences.

Line 25: The beginning of this phrase contradicts the general idea of the manuscript. We do not know for sure if it will have these great effects. Therefore, the authors should include the word “potentially” after “It will”.

Lines 34-39: I think we have a circular problem here. If the society do not understand the benefits brought by science, they will not change behavior. If the behavior is not changed by the society, the applications and benefits of scientific discoveries may be impaired. One cannot expect a change in societal behavior if they are not convinced by the potential benefits. The authors could include that the changes in citizen behavior depends on the success of accurate and clear science divulgation – as proposed in the manuscript.

Line 50: I suggest replacing the word “will” by “can”, for the same reason told in a comment above.

Line 54: Values of many people are not linked to science. “Personal benefits” would fit better here.

Line 58: Please also write benefit in plural, because it can be more than one.

Lines 60-62: I agree that it helps to prevent, but not always. Sometimes non-scientific arguments raise in the society against a new technology, even if the science is well explained. Religious arguments are the most common, mainly in developing countries. Then the phrase would be more accurate if replacing “prevents” by “helps prevent”.

Lines 71-83: I suggest adding letters or numbers to the beginning of each phrase in this part. Also, replacing the “,” by “;” in the end of each phrase will make it sounds better.

Line 73: Please review this phrase, “are have been” is not correct.

Line 86: Technical knowledge “that” scientists…

Lines 86-87: This sentence needs to be clarified. What the authors mean with “…it is also aligned with personal values and politics”? True science needs to be absent of ideology or it tends to be biased. If the authors are stating here that scientists need to be ideologically engaged with a cause to impact society, I strongly disagree. It will actually contribute to the overselling of results, which is criticized in this manuscript. Science tests hypothesis and disclose the results impartially. The impact science will have on society depends on science divulgation, which needs to be performed in alternative ways in addition to the actual results published in scientific articles – which contains a highly technical wording that is not understandable by the general public but is very important to the scientific community in that field. The authors should acknowledge that the current method of scientific literature is very important for science development, but other complementary media could help science divulgation to impact society, e.g. books, magazines, social media, using more appropriate wording for the general public, with the methods suggested in this manuscript.

Lines 87-93: This part needs changes. The interdisciplinary approach suggested is interesting and potentially can generate a greater impact in society. However, the current method of scientific literature (articles in peer reviewed journals) cannot be shifted towards an excessive practical reason. Freedom of thinking and curiosity has always been the main fuel that stimulates scientists and scientific production. Therefore, I disagree with the authors on this statement: “what people need to know, rather than what may appeal to public or what scientists want to disclose”. What scientists want to disclose should be conserved in the technical scientific literature. The word “intersection” also makes no sense in that phrase. Nevertheless, I agree with the authors that the proposed interdisciplinary approach might be useful for science divulgation to the general public using alternative media, as mentioned in the comment above. Following this idea, scientists could start to also write in these alternative media with the methods proposed here (interdisciplinary approach, clear and accurate information containing potential benefits and limitations, focused in more practical subjects of general interest) in order to improve the impact of microbiome research in society.

Lines 96-97: There is an acknowledgement? Then please add a reference for this. Or clearly state that this is an opinion of the authors.

Lines 97-100: I understand the idea, but there are two different things that should be separated here. The first is the microbiome science. The society can interact more with scientists to provide ideas to be tested, for example. But you need to be a scientist to make science. It is impossible to the general public be protagonists in this process because they lack the knowledge about the premises and processes of science. The second is the microbiome applications. In this case, scientists can suggest directions and invest on science divulgation (as proposed here) to improve the benefits of microbiome knowledge, but the decision of its applications is made by the society. Scientists have little power on that. The success of use in the industry and farmers depends on how they agree with the knowledge/technology. The policymakers that can increase microbiome applications are also elected by society and scientists have a reduced power on that.

Box 1: The benefits for the consumers should be clarified and amplified in the message. All others are good.

Line 131: Please correct “focused”.

Table 1: Please see my first comments and update where (environments) microbiomes are found in this table.

Lines 138-143: Again, numbers or letters would help to organize this text.

Line 154: I do not understand to whom “[your messages]” is destined. To the scientists that want to participate in the CIRCLES project? Please clarify.

Lines 174-176: Please review this phrase, there is something missing to make sense.

Line 217: Again, take care with this statement. What scientists want to communicate is important as well. Who knows what is really important? Important to whom? Depends on the referential. Make clear you refer exclusively to the communication in alternative media aiming to increase the divulgation of microbiome science to the general public.

Line 225: Please review grammar in this phrase.

Lines 243 and 252: How scientists would be benefited from this process was not properly explored. Visibility is not enough, the authors need to propose and discuss practical and tangible benefits that scientists would have in joining this initiative of increasing microbiome science divulgation. What is the role of the academic/research institutions in this process?

Reviewer 3 Report

The paper by Schelkle and Galland deals with the topic of how to propagate knowledge of the food (?) microbiome to the broader public. This is not a research paper, nor a review paper. I’m having trouble understanding why it was submitted to this particular journal. It’s about the microbiome or so the authors want us to believe. However, this is just as much about explaining science oriented issues to non-scientists. This has already been better elaborated on in other papers or educational forums. In this regard I doubt that this paper will provoke any feedback once it is published. It simply lacks substance. Its just a vague, wordy “encouragement” to promote scientific issue awareness without any solid guidelines. I’m sorry but this paper didn’t provide me any insight.
